# Quantitative Ethnobotanical Analysis of Medicinal Plants of High-Temperature Areas of Southern Punjab, Pakistan

**DOI:** 10.3390/plants10101974

**Published:** 2021-09-22

**Authors:** Muhammad Usman, Allah Ditta, Faridah Hanum Ibrahim, Ghulam Murtaza, Muhammad Nawaz Rajpar, Sajid Mehmood, Mohd Nazre Bin Saleh, Muhammad Imtiaz, Seemab Akram, Waseem Razzaq Khan

**Affiliations:** 1Department of Botany, Government College University, Lahore 54000, Punjab, Pakistan; usmanphytologist@gmail.com; 2Department of Environmental Sciences, Shaheed Benazir Bhutto University Sheringal, Upper Dir 18000, Khyber Pakhtunkhwa, Pakistan; 3School of Biological Sciences, The University of Western Australia, Perth, WA 6009, Australia; 4Institute Ekosains Borneo, Universiti Putra Malaysia Kampus Bintulu, Bintulu 97008, Malaysia; f_hanum@upm.edu.my; 5Faculty of Environmental Science and Engineering, Kunming University of Science and Technology, Kunming 650500, China; murtazabotanist@gmail.com; 6Department of Forestry, Faculty of Life Sciences, SBBU Sheringal, Upper Dir 18000, Khyber Pakhtunkhwa, Pakistan; rajparnawaz@gmail.com; 7College of Environment and Ecology, Hainan University, Haikou 570100, China; drsajid@gzhu.edu.cn; 8Department of Forestry Science and Biodiversity, Faculty of Forestry and Environment, Universiti Putra Malaysia, Sri Serdang 43400, Malaysia; nazre@upm.edu.my; 9Soil and Environmental Biotechnology Division, National Institute for Biotechnology and Genetic Engineering, Faisalabad 38000, Punjab, Pakistan; m.imtiazpk92@hotmail.com; 10Department of Biology, Faculty of Science, University Putra Malaysia, Seri Kembangan 43400, Malaysia; seemabakram@ymail.com

**Keywords:** ethnobotanical survey, ethnomedicinal plants, *Poaceae*, cardiovascular diseases, healers, informant consensus factor

## Abstract

Lack of proper infrastructure and the poor economic conditions of rural communities make them dependent on herbal medicines. Thus, there is a need to obtain and conserve the historic and traditional knowledge about the medicinal importance of different plants found in different areas of the world. In this regard, a field study was conducted to document the medicinal importance of local plants commonly used by the inhabitants of very old historic villages in Southern Punjab, Pakistan. In total, 58 plant species were explored, which belonged to 28 taxonomic families, as informed by 200 experienced respondents in the study area. The vernacular name, voucher number, plant parts used, and medicinal values were also documented for each species. Among the documented species, *Poaceae* remained the most predominant family, followed by *Solanaceae* and *Asteraceae*. The local communities were dependent on medicinal plants for daily curing of several ailments, including asthma, common cold, sore throat, fever, cardiovascular diseases, and digestive disorders. Among the reported species, leaves and the whole plant remained the most commonly utilized plant parts, while extracts (38.8%) and pastes (23.9%) were the most popular modes of utilization. Based on the ICF value, the highest value was accounted for wound healing (0.87), followed by skincare, nails, hair, and teeth disorders (0.85). The highest RFC value was represented by *Acacia nilotica* and *Triticum aestivum* (0.95 each), followed by *Azadirachta indica* (0.91). The highest UV was represented by *Conyza canadensis* and *Cuscuta reflexa* (0.58 each), followed by *Xanthium strumarium* (0.37). As far as FL was concerned, the highest value was recorded in the case of *Azadirachta indica* (93.4%) for blood purification and *Acacia nilotica* (91.1%) for sexual disorders. In conclusion, the local inhabitants primarily focus on medicinal plants for the treatment of different diseases in the very old historic villages of Southern Punjab, Pakistan. Moreover, there were various plants in the study area that have great ethnobotanical potential to treat various diseases, as revealed through different indices.

## 1. Introduction

Over the past few decades, several studies have indicated that the use of medicinal plants to treat various ailments is a common trend around the world [1,2,3]. Ethnobotanical surveys are often conducted to evaluate the complex connection between local communities and wild species of plants [4]. These surveys are crucial in understanding the cultural believes associated with the use and value of native plants. These types of studies help scientists to discover some novel drugs from plants [5]. Furthermore, ethnobotanical surveys are confirming the significance of medicinal plants from a socioeconomic perspective [6,7]. These sorts of surveys are important in preserving the indigenous knowledge about medicinal plants of a particular area [8,9]. These surveys provide a window to discover some new active compounds found in the plants against some deadly diseases [7]. Traditional ethnobotanical data on medicinal plants are required for the protection, conservation, and development of herbal drugs [2,10,11]. 

Since ancient times, humans have started extracting and processing valuable materials from medicinal plant species to cure and treat several diseases [11]. Medicinal plants play a significant role in uplifting the economic conditions and living standards of the local communities in remote areas [12]. There are more than 20,000 wild plant species in the world, while not more than twenty plant species fulfill 90% of the world’s food requirements. The use of medicinal plants is a common practice predominantly in the developing countries of the world. Some regions in Asian countries are known to use the whole plant or plant parts for the treatment of several ailments [13]. In this modern era, approximately 25% of the drug’s sources are from plants. It is expected that this trend would increase in the upcoming years [14,15]. However, there are certain areas of the world that have not been surveyed to collect and preserve the indigenous knowledge about medicinal plants. Traditional knowledge is often held by elderly people and traditional healers; unfortunately, the younger generations are not focused on obtaining this valuable knowledge from them [16]. The modern healthcare system is also imposing problems for traditional practices to cure diseases using medicinal plants [17].

According to World Health Organization (WHO), approximately 65% of the world population is dependent on traditional practices as primary health care [14,18]. Modern medicines are too expensive, especially for the people living in underdeveloped or even in developing countries, so they always trust in traditional practices employing local plants until reaching a critical situation [19].

Pakistan has a great diversity of medicinal plants in different ecological zones. The country has over 600 wild plant species that are medicinally valuable [20]. In Pakistan, most of the population is dependent on herbal drugs, apart from some big cities [2,12,21,22]. As mentioned earlier, the traditional knowledge about the ethnomedicinal importance of plants in different areas of the country needs to be preserved for our future generations. In addition, this knowledge could be used for the development of new medicines. To our knowledge, no study has been conducted to elucidate the ethnomedicinal potential of local plants in the present study area. Therefore, this study was conducted to compile the traditional ethnobotanical knowledge about different plants in four villages in Khanewal, Punjab, Pakistan. 

## 2. Materials and Methods 

### 2.1. Study Area 

The present study was conducted in the historically old villages of Tehsil Kabirwala, District Khanewal, Southern Punjab, Pakistan (Figure 1). These villages are comparatively remote and situated near the bank of River Ravi. Moreover, these villages do not have modern healthcare centers and people for centuries are relying on medicinal plants. Unfortunately, no research work has been conducted to document the ethnomedicinal knowledge of plants found in these villages in the past. The soils in these study areas are fertile, suitable for wheat and cotton production. The climate is dry and extremely hot; light rain leaves the land dusty and arid while temperature range from 30.8 to 47 °C in summer, whereas in winter, it ranges from 15.6 to 25.7 °C [23]. As far as the language is concerned, approximately 90% of the population of these villages speak the Saraiki language. The Saraiki language is majorly spoken in Southern Punjab, Pakistan. This language has emerged from several dialects after the independence of Pakistan in 1947. Saraikstan is a combined region combined of different areas where this language is spoken. Saraiki language people are considered friendly and known for their hospitality. Some 40,000 years back, the Saraiki region became part of the Indus Valley Civilization. About 8.38% of the people in Pakistan speak the Saraiki language. Most of the Saraiki language people are living in the rural areas and healthcare centers are far away from their reach. Therefore, these people have remained more inclined towards the use of medicinal plants for centuries [24]. This area is famous for mango production, while other agricultural products include citrus.

### 2.2. Field Surveys and Data Collection 

Fieldwork was carried out during the course of approximately two years, from January 2016 to July 2018. Six field tours were arranged to collect the best possible ethnobotanical data by simple interviewing or using structural questionnaires. Four villages were selected, namely, Kot Islam, Haveli Koranga, Qatal Pur, and Sarai Sidhu. Before the conduction of a formal interview, verbal consent to participate in this research work was given by all the participants. In total, 200 people were interviewed, including 125 (62.5%) males, 74 (37.0%) females, and 1 (0.5%) other. The majority of the selected respondents were old-aged because they usually have more knowledge and expertise in using medicinal plants for particular disorders. Educational background of the respondents was as follows: illiterate 171 (85.5%), primary 12 (6.0%), middle 7 (3.5%), secondary 5 (2.5%), higher secondary 3 (1.5%), and university 2 (1.0%). Ethnobotanical knowledge usually passes from generation to generation verbally and most of the people from Pakistan have expertise in using medicinal plants despite the high rate of illiteracy. It is a common trend that most of the older people will not even write a single sentence, but they are genius in using medicinal plants and are famous for their knowledge about the medicinal plants in the surrounding areas. Therefore, these people have a bundle of knowledge about the use of medicinal plants despite being illiterate. In the case of illiterate healers, the basic information and other study-related questions were recorded by the interviewer during the conversation with the respondents. Interviews were conducted mainly during the evening hours when the respondents were usually free from their daily routine work.

Moreover, common places such as hamlets, tea stalls, farms, and gardens were selected to conduct interviews. To obtain detailed information, the authors visited the farms and houses at least twice or thrice to engage in conversation, especially with the old age people whose ages ranged from 70 to 98 years. Furthermore, the local community was informed about conducting interviews and seeking permission before interviewing every individual. During interviews, the demographic characteristics of each respondent and vernacular names of the plants, plant parts used as medicine, and the preparatory methods of the plants were also documented. 

### 2.3. Inventory 

Plants were collected, and all the necessary data were documented, including their vernacular names, locality, and medicinal values. Plants were identified with the assistance of taxonomic experts available at the Department of the Botany, Government College University, Lahore, Pakistan and with the available literature present in the Science Library of the Government College University, Lahore, Pakistan [25,26]. Moreover, the plants were also compared with the existing plants available at the university’s herbarium, the Sultan Ahmad Herbarium, Government College University, Lahore, Pakistan. The collected specimens were pressed, dried out, and deposited at the Sultan Ahmad Herbarium, Government College University, Lahore, Pakistan.

### 2.4. Ethical Consideration

Data collection was carried out after ethical considerations to avoid or minimize anything that can cause emotional or physical damage to the respondents. Voluntary participation was followed to collect ethnobotanical data from only those respondents who were excited and willing to share their experience regarding the use of medicinal plants.

### 2.5. Quantitative Ethnobotany

#### 2.5.1. Informant Consensus Factor

Informant consensus factor (ICF) was calculated by following Heinrich et al. [27]:(1)$$\mathrm{ICF}=\frac{N_{\mathrm{ur}}-N_{t}}{N_{\mathrm{ur}}-1}$$
where “N_ur_” is the number of use citations for each disease category and “N_t_” represents the number of taxa used for that disease category. The homogeneity of the information regarding a particular category of ailment is highlighted by the ICF values [18,28].

#### 2.5.2. Use Value

The use value (UV) represents the relative importance of a particular plant. Moreover, higher UVs indicate that there are many use reports for a given plant species, whereas the value near zero indicates fewer reports, confirming the use of the species. It was calculated by using the following formula [29,30]:(2)$$\mathrm{UV}=\frac{\sum^{\;}U_{i}}{N}$$
where “∑U_i_” represents the total number of uses cited by each informant for a given species, while “N” refers to the total number of respondents or informants who participated in the survey [31]. 

#### 2.5.3. Relative Frequency Citation

The relative frequency citation (RFC) was calculated by using the formula as follows [18,32];
(3)$$\mathrm{RFC}=\frac{\mathrm{FC}}{N}$$
where FC indicates the number of respondents or informants who mentioned the use of the species while “N” represents the total number of respondents or informants involved in the study.

#### 2.5.4. Fidelity Level

The fidelity level (FL) indicates the preference of one species over others for the treatment of a particular ailment [32,33]. For the calculation of the FL of plant species, the following formula was used:(4)$$\mathrm{FL}\;\left(\%\right)=\frac{I_{p}}{I_{u}}\times 100$$
where I_p_ is the number of respondents who mention the use of species for a particular ailment category, while “I_u_” is the number of respondents who cited the use of that species for any ailment category.

### 2.6. Data Analysis

The collected data were subjected to quantitative analysis using Microsoft Excel 2016 regarding the percentages and graphical presentation. Arc-GIS version 10.7 was used to make a study area map

## 3. Results and Discussion 

### 3.1. Demographic Characteristics of Respondents 

Six field tours were arranged in different seasons during the course of two years (2016–2018) to collect and document the best possible ethnobotanical data. Two hundred respondents were interviewed, and the demographic characteristics of each respondent are documented in Table 1. Among the selected respondents, 62.5% were males, 37.0% were females, and others were 0.5%. Most of the respondents were illiterate (85.5% of the respondents), while only 1% had a university education. As far as marital status was concerned, only about 13% were unmarried. Most of the respondents spoke the Saraiki language fluently, while Urdu-speaking respondents were less common.

### 3.2. Health Issues 

The data showed that the local inhabitants were suffering from several health issues, the most common being urinary disorders, gastrointestinal (GIT) disorders, respiratory disorders, sexual disorders, muscular disorders, cardiac disorders, skin and eyes problems, ulcer, diabetes, and snake or mosquito bites (Figure 2). Several studies have shown that nearly all these health issues were common in different parts of the country [1,2,19,21,22,34,35].

### 3.3. Medicinal Plant Diversity

During fieldwork, about 58 plants were documented for medicinal purposes (Table 2). The collected 58 plant species belonged to 28 taxonomic plant families. The *Poaceae* family, with six medicinal species, was the most dominant, followed by Asteraceae and Solanaceae with five species each. The *Brassicaceae* and *Fabaceae* families represented four species each; *Lamiaceae* and *Moraceae* represented three species each. Furthermore, other frequently used plants belonged to families such as *Amaranthaceae*, *Amaryllidaceae*, *Apiaceae*, *Chenopodiaceae*, *Convolvulaceae*, *Meliaceae*, and *Myrtaceae*, with two species each. Finally, the *Anacardiaceae*, *Rutaceae*, *Primulaceae*, *Rhamnaceae*, *Salvadoraceae*, *Oxalidaceae*, *Moringaceae*, *Lythraceae*, *Euphorbiaceae*, *Cucurbitaceae*, *Combretaceae*, *Boraginaceae*, *Asphodelaceae*, and *Apocynaceae* families had one medicinal species each.

### 3.4. Medicinal Plant Parts Used for the Treatment of Diseases

The leaves were the most commonly used plant parts by the local inhabitants of the study areas. Leaves were reported 33 times in the documented 58 plants followed by the use of the whole plant (21); flowers and fruits were cited by respondents 15 times each (Figure 3). Moreover, other plant parts used to treat different diseases included bark (14 times), roots (12 times), seeds (10 times), and stem or shoot (7 times). Leaves can easily be collected and require less effort in using them to treat different diseases compared to the other plant parts i.e., flowers, roots, fruits, and seeds. Earlier, Giday et al. [36] reported that leaves were the most frequently used plant parts to treat various ailments due to less effort required for their use. According to Kadir et al. [37], leaves removed from the plants cause less damage compared to those observed with the removal of other plant parts. The removal of other plant parts can severely affect plant growth and development.

On the other hand, several studies have shown that the leaves contain more bioactive compounds against particular diseases and play a major role in the preparation of herbal medicines [2,12,34,36,38,39]. In contrary to the present study, whole plant parts were frequently used to treat common diseases in other parts of the country, predominantly in the Khyber Pakhtunkhwa province [40].

### 3.5. Mode of Utilization 

As is clear in Figure 4, the medicinal plants found in the study areas were most commonly used in the form of an extract (38.8%), paste (23.9%), and decoction (11.9%). All these forms of utilization are pretty common in different regions of the country and the world [2,7,12,21,34,41,42]. Most of the plants were used internally while some were also used externally in the form of paste or powder. For the preparation of different herbal products, the respondents in the study areas preferred fresh plant materials compared to dried ones.

### 3.6. Number of Taxa Curing or Treating Ailment Categories

About 14 different ailment categories were determined from the number of taxa used for the treatment of these ailment categories (Figure 5). Cough, fever, cold, and flu were among the ailment categories treated by the maximum number of taxa (29), followed by GIT disorders (27) and skincare (26). Furthermore, only four taxa were reported for the treatment of cardiac disorders. The results of the current study have shown that the majority of respondents perform their work in the fields and are more prone to fever, cold, and flu; therefore, they were simply aware of the use of medicinal plants for the treatment of these health issues.

### 3.7. Quantitative Data Analysis 

#### 3.7.1. Informant Consensus Factor

In the present study, the informant consensus factor (ICF) was examined for 14 ailment categories: (1) GIT disorders; (2) urinary disorders; (3) respiratory disorders; (4) sexual problems; (5) snake or mosquito bite; (6) wound healing; (7) blood purification; (8) cough, cold, fever, and flu; (9) cardiac disorders; (10) diabetes; (11) skincare and nails, hair, and teeth disorders; (12) muscular disorders or rheumatism; (13) ulcers; and (14) eye problems. The ICF value ranged from 0.60 to 0.87, while the average ICF value was 0.74 (Table 3). The highest value of ICF was reported for wound healing (0.87), followed by skincare and nails, hair, and teeth disorders (0.85), and then GIT disorders (0.83). The wounds are the most common ailment in the study area, which might be due to the low literacy level. The local people used their fields to grow crops for the fulfillment of their household requirements and it might be the reason that they were more prone to injury while working in the fields [21,26]. On the other hand, when people work in the fields during hot summer days, they ultimately suffer from GIT disorders due to extreme heat. Therefore, the highest value was reported for wound healing, teeth disorders, and gastrointestinal disorders. These results are nearly parallel to the results of the study carried out by Zahoor et al. [2]. The results of the current study have shown that the local inhabitants were using 14 different species for wound healing; the most cited plants were *Lawsonia inermis*, *Acacia nilotica*, and *Ziziphus mauritiana*. Towfik et al. [43] reported the higher use of *L. Inermis* for wound healing in the hot and dry regions of Saudi Arabia. The farmers are injured during their fieldwork and use *Acacia nilotica* for swift wound healing [44].

#### 3.7.2. Use Value

In the present study, the use value (UV) ranged from 0.032 to 0.58 (Table 4). The highest UVs were represented by *Conyza canadensis* (0.58) and *Cuscuta reflexa* (0.58), followed by *Xanthium strumarium* (0.37). The lowest UVs were observed in the case of *Withania somnifera* (0.032). Several studies have also found similar plants as reported in the present study with the highest UVs in other parts of the country [2]. The UV shows the relative importance of the use of medicinal plant species in a specific area [45]. However, it is important to understand that there is a great diversity of plants and environmental fluctuations in different parts of Pakistan. Therefore, local inhabitants often exploit the local and easily available plants to treat common territorial diseases. There will always remain a slight difference in the use value of plants even in two neighboring districts of the same country.

#### 3.7.3. Relative Frequency Citation

In the present study, the value of the relative frequency citation (RFC) ranged from 0.04 to 0.95 (Table 4). As is clear from the data, the highest value of RFC was recorded in the case of the native plants of the area. The highest value was found in the case of *Acacia nilotica* (0.95) and *Triticum aestivum* (0.95), followed by *Azadirachta indica* (0.91), *Ficus benghalensis* (0.89), and the lowest RFC in the case of *Chloris virgata* (0.04). The local inhabitants are well aware of the use of locally available medicinal plants [2,34]. Earlier, it has been found that *Acacia nilotica* and *Azadirachta indica* preferably grow under high-temperature conditions [46]. Moreover, *Triticum aestivum* is commonly grown as a cash crop in the present areas of the country [23].

#### 3.7.4. Fidelity Level

The value of the fidelity level (FL) ranged from 25.0 to 93.4% (Table 4). It is a fact that the higher the value of FL, the higher will be the plant’s usage [18]. In the present study, the highest FL was represented by *Azadirachta indica* (93.4%) for blood purification followed by *Acacia nilotica* (91.0%) for sexual disorders, and *Lawsonia inermis* (89.4%) for ornamentation of the body (hair dye, staining of hair, hands, legs, and nails). At the same time, the lowest FL was recorded in the case of *Chloris virgata* (25.0%) for fodder. 

Medicinal plants are gaining interest over the years predominantly in Asian countries due to their role in the provision of food, healthcare support, and poverty alleviation. Nine Asian countries’ researchers, namely, China, Korea, India, Indonesia, Malaysia, Myanmar, Sri Lanka, Thailand, and Vietnam, are more focusing on the discovery of novel drugs from plant material. All these nine countries have already published their National Monographs of herbal drugs [47]. Moreover, other Asian countries are also investing more energy and money in the discovery of herbal drugs. Asian communities are experts in the usage of medicinal plants for the treatment of common local diseases [3]. Interestingly, both men and women are well informed from their ancestors regarding traditional treatment with medicinal plants. Undeniably, their expertise is owned by their ancestors as they have received this valuable knowledge from their ancestors [48]. The present study suggests that there is a dire need to increase the scientific investigation for the discovery of bioactive compounds from medicinal plants. These types of surveys assist national and international researchers to investigate the medicinal plants for the synthesis of novel drugs [49,50]. 

Several studies have confirmed the extensive use of medicinal plants as observed in the present study [42,51,52]. A single plant or parts of a plant species can be used for traditional herbal medicines [22,52]; however, several reports have indicated that a single herbal recipe for particular diseases can be prepared using different plant species [3]. The therapeutic plants are vital in treating diseases of about 80% of the people from emerging economies [53,54]. Traditional healers (Hakeems) are always the vital part of these communities, which are more dependent on medicinal plants given the poor healthcare system in such areas. In the northern regions of the country, the local inhabitants collect therapeutic plants and sell them in the market to meet their daily household financial requirements [45]. It is important to understand that the majority of plants used in the study area for the treatment of diseases are easily available and people know how to use these plants sustainably.

### 3.8. Ethnopharmacological Relevance

The present research work is novel compared to ethnomedicinal literature of other areas of Southern Punjab [3,55,56]. The results only have been compared to ethnomedicinal literature of neighboring areas because the distant areas have fewer similarities. Comparative studies have revealed that twenty plant species, including *Albizia lebbeck*, *Amaranthus spinosus*, *Brassica nigra*, *Chenopodium murale*, *Conyza Canadensis*, *Cordia dichotoma*, *Coronopus didymus*, *Cucurbita pepo*, *Eclipta alba*, *Eclipta prostrate*, *Eucalyptus globules*, *Hordeum vulgare*, *Lawsonia inermis*, *Ocimum basilicum*, *Ocimum sanctum*, *Ricinus communis*, *Salvadora oleoides*, *Sisymbrium irio*, *Sonchus arvensis*, and *Terminalia arjuna*, have not been documented previously. In the present study, newly reported species with their most common uses for the first time include *Albizia lebbeck* (diarrhea), *Amaranthus spinosus* (snakebite), *Brassica nigra* (rheumatism), *Chenopodium murale* (blood purification), *Conyza Canadensis* (diarrhea), *Cordia dichotoma* (anti-inflammatory), *Coronopus didymus* (malaria), *Cucurbita pepo* (vermifuge), *Eclipta alba* (diabetes), *Eclipta prostrate* (hair dye), *Eucalyptus globules* (wound healing), *Hordeum vulgare* (cough), *Lawsonia Inermis* (hair dye, staining of hair, hands, legs, and nails), *Ocimum basilicum* (menstrual cramps), *Ocimum sanctum* (rheumatism), *Ricinus communis* (laxative), *Salvadora oleoides* (stomach imbalance), *Sisymbrium irio* (wound healing), *Sonchus arvensis* (asthma), and *Terminalia arjuna* (diabetes). 

The present study also reported that nineteen species could be used for the treatment of rheumatism. In the literature, no study has reported such a good number of species for the treatment of rheumatism. On the other hand, sixteen species have been documented for the treatment of diarrhea and these results are pivotal for the researchers to focus on the detection of useful bioactive compounds for the preparation of novel drugs.

Use of the traditional medicinal plants for the treatment of different diseases has actively been emerging in the recent decade. Various researchers have found that ethnopharmacological relevance can be examined through comparison with previous studies. *Amaranthus spinosus* is used to treat diarrhea, fever, snakebite, and fodder [2,57]. *Chenopodium album* is another widely distributed plant species in the country, used against rheumatism, as a diuretic, and for cooking purposes [40,58]. *Ricinus communis* is used as a laxative and for the treatment of GIT problems [2,59]. *Dalbergia sissoo* is used for sexual disorders, including vaginal infection, sperm production, and used as a diuretic. The use of this particular species for sexual disorders coincides with a particular study carried out in India [40,60]. *Acacia nilotica* is a well-known therapeutic plant of the Asian subcontinent, used for the treatment of sexual disorders in both sexes, wound healing, and GIT disorders (diarrhea) [12,60,61]. *Azadirachta indica* is another plant used for the treatment of several diseases, including skincare, hair problems, birth control, abortion, and diabetes [2,12,62]. *Ficus benghalensis* is commonly used to treat diarrhea, jaundice, diabetes, vaginal infection, and is known to possess anti-inflammatory activities [2,63]. *Morus alba* can be used for the treatment of cough, common cold, and sore throat [64]. *Ficus religiosa* plant extract and paste is used for the treatment of ulcer and wound healing [40]. *Allium cepa* is commonly used in cooking across the globe; however, it exhibits great medicinal properties. It is used to treat sexual disorders in sexes, blood purification, sore throat, hair loss, appetite problems in cattle, wound healing, and regulation of normal blood pressure [65,66].

## 4. Conclusions

The collected data showed that the local community was dependent on herbal drugs. The majority of the population relied on the local herbalists or used medicinal plants using their ethnobotanical knowledge due to the extra financial burden posed by the purchase of modern medicines. During the field survey, more focus was on older people, to obtain the maximum and more valuable ethnobotanical knowledge about medicinal plants. Unfortunately, the young generation is not focusing and trying to obtain or carry on this traditional knowledge due to modernization. Another possible reason is that there is a gap between older people and the young generation due to modern entertainment tools. The present study found that the *Poaceae* (6), *Asteraceae* (5), and *Solanaceae* (5) families represented the maximum number of medicinal plants. Different quantitative indices, such as ICF, UV, RFC, and FL, indicated that the local community in the study area was dependent on herbal drugs because of their poor economic conditions and lack of proper healthcare centers. Based on the ICF values, the highest value was found for wound healing (0.87), followed by skincare, nails, hair, and teeth disorders (0.85). The highest RFC value was represented by *Acacia nilotica* and *Triticum aestivum* (0.95 each), followed by *Azadirachta indica* (0.91). The highest UV was represented by *Conyza canadensis* and *Cuscuta reflexa* (0.58 each), followed by *Xanthium strumarium* (0.37). As far as FL was concerned, the highest value was recorded in the case of *Azadirachta indica* (93.4%) for blood purification and *Acacia nilotica* (91.1%) for sexual disorders. In conclusion, there were various plants found in the study area, which have great ethnobotanical potential to treat various diseases, as revealed through different indices. In the future, it is expected to further identify species of medicinal plants potentially useful to humans in the same study area, with the help of field botanists [67,68,69,70,71] who have recently gained the attention of the global scientific community.

## Figures and Tables

**Figure 1 plants-10-01974-f001:**
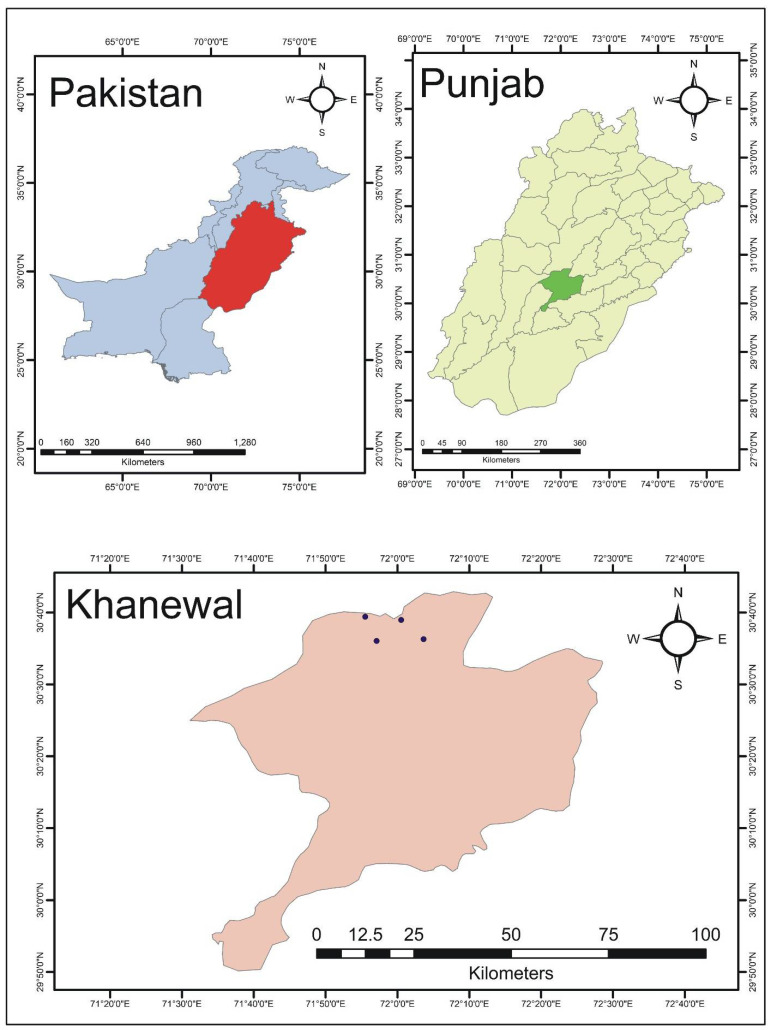
Map of the study area. Note: The dots show the exact location of the villages.

**Figure 2 plants-10-01974-f002:**
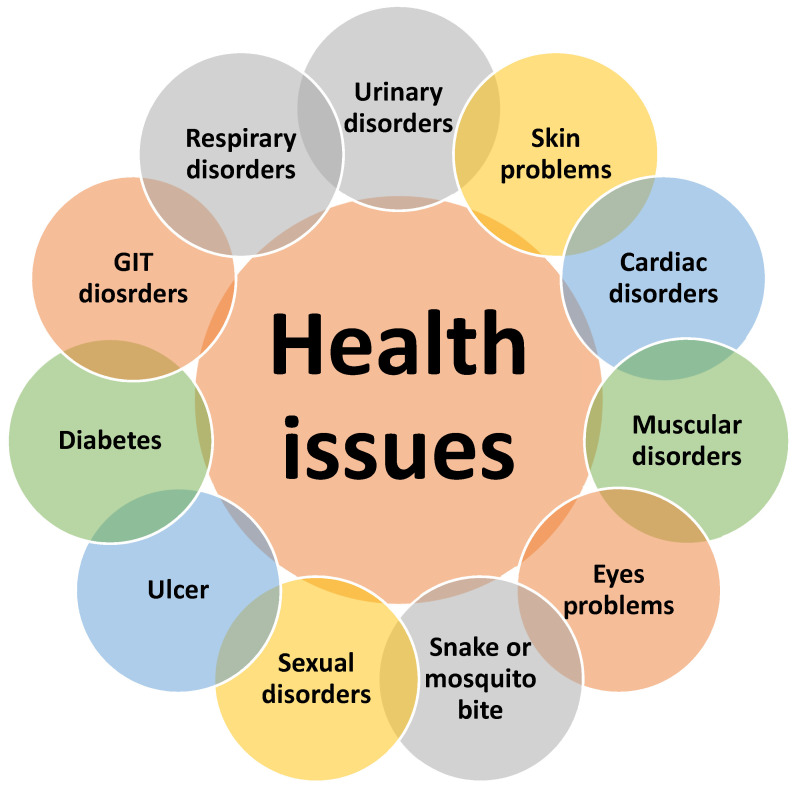
Health issues of the people in the study areas.

**Figure 3 plants-10-01974-f003:**
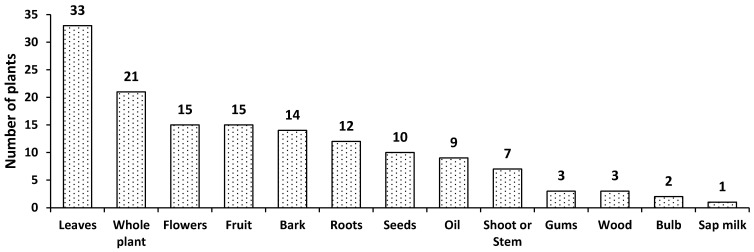
Plant parts used.

**Figure 4 plants-10-01974-f004:**
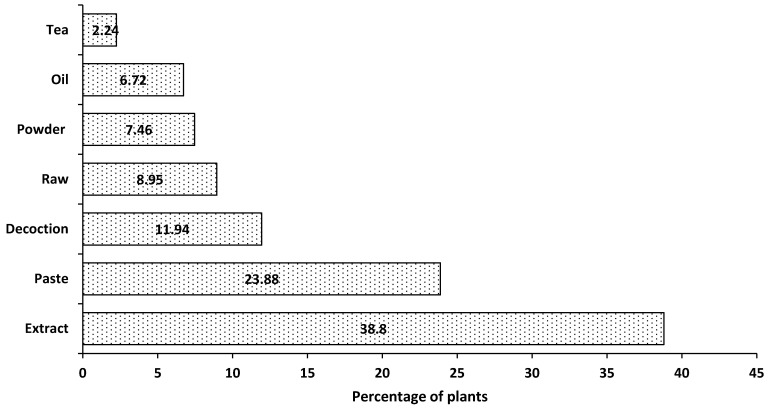
Mode of administration.

**Figure 5 plants-10-01974-f005:**
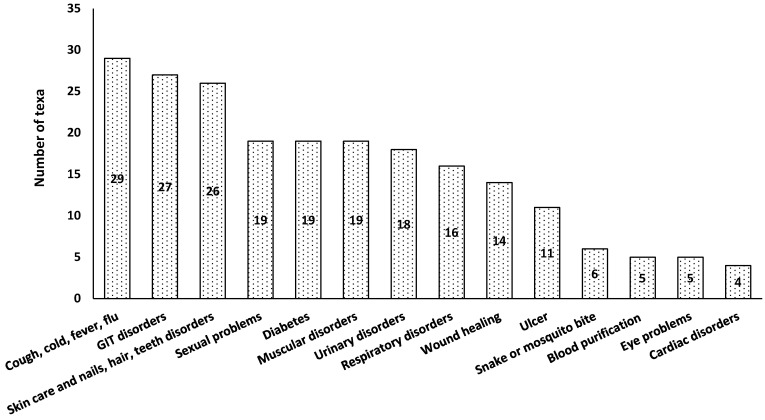
The number of taxa used for treating different ailments.

**Table 1 plants-10-01974-t001:** Demographic characteristics of the informants.

Variation	Category	Number	Percentage
Gender	Male	125	62.5%
	Female	74	37.0%
	Other	01	0.5%
Age	Less than 30	11	5.5%
	30–40	15	7.5%
	40–50	32	16.0%
	50–60	37	18.5%
	60–70	43	21.5%
	Over 70	62	31.0%
Marital status	Unmarried	26	13.0%
	Married	174	87.0%
Educational background	Illiterate	171	85.5%
	Primary	12	6.0%
	Middle	07	3.5%
	Secondary	05	2.5%
	Higher secondary	03	1.5%
	University	02	1.0%

**Table 2 plants-10-01974-t002:** Families representing the number of medicinal plants.

Sr. No.	Family	Number of Medicinal Plants
1	Poaceae	6
2	Asteraceae	5
3	Solanaceae	5
4	Brassicaceae	4
5	Fabaceae	4
6	Lamiaceae	3
7	Moraceae	3
8	Amaranthaceae	2
9	Amaryllidaceae	2
10	Apiaceae	2
11	Chenopodiaceae	2
12	Convolvulaceae	2
13	Meliaceae	2
14	Myrtaceae	2
15	Anacardiaceae	1
16	Asphodelaceae	1
17	Apocynaceae	1
18	Boraginaceae	1
19	Combretaceae	1
20	Cucurbitaceae	1
21	Euphorbiaceae	1
22	Lythraceae	1
23	Moringaceae	1
24	Oxalidaceae	1
25	Primulaceae	1
26	Rhamnaceae	1
27	Rutaceae	1
28	Salvadoraceae	1

**Table 3 plants-10-01974-t003:** ICF values of the medicinal plant species exploited for the treatment of several ailment categories.

Ailment Categories	N_ur_	N_t_	ICF
Wound healing	103	14	0.87
Skincare and nails, hair, and teeth disorders	167	26	0.85
GIT disorders (Diarrhea, constipation, stomach imbalance, cholera)	156	27	0.83
Muscular disorders (Rheumatism)	89	19	0.79
Diabetes	78	19	0.77
Eye problems	17	05	0.75
Cardiac disorders	13	04	0.75
Blood purification	16	05	0.73
Ulcer	37	11	0.72
Urinary disorders	57	18	0.70
Cough, cold, fever, flu	89	29	0.68
Sexual problems	58	19	0.68
Snake or mosquito bite	14	06	0.61
Respiratory disorders (asthma, tuberculosis)	39	16	0.60

Note: N_ur_—number of use citations for each disease category; N_t_—number of taxa used for that disease category; ICF—informant consensus factor.

**Table 4 plants-10-01974-t004:** Plants from the study areas with medicinal values.

S. No	Plant species with Specimen Name	Family	Vernacular Name	Plant Parts Used	Methods of Utilizations	Uses	UV	RFC	FL%
1	*Acacia nilotica* (L.) Willd. ex Del. JP–454 *	Mimosaceae	Kekar	Leaves, roots, fruit, flowers, bark, gum, oil	Extract, decoction, powder	High blood pressure, liver problems, tuberculosis, **sexual disorders**, ulcer, wounds, common cold, diarrhea, toothache, pickle, fodder	0.058	0.95	91.1
2	*Albizia lebbeck* (L.) Benth. JP–414	Fabaceae	sharaii	Leaves, flowers, bark,	Paste	Lungs disorders, diabetes, anti-inflammatory, cough, flu, eye care, **diarrhea**, piles	0.11	0.35	60.0
3	*Allium cepa* L. JP–421	Amaryllidaceae	Ganda/Wasal	Bulb, shoots, flower	Extract, raw, decoction, paste	Diabetes, high blood pressure, asthma, **sexual disorders** in both sexes, dry cough, sore throat, hair loss, sleeplessness, appetite problems in cattle, wounds, cooking	0.062	0.88	60.2
4	*Allium sativum* L JP–379	Amaryllidaceae	Thoum/Lehson	Bulb, leaves, flower	Extract, decoction, paste, raw,	Diabetes, High blood pressure, cardiac disorders, cough, fever, **toothache**, skin problems, tuberculosis, cooking	0.058	0.78	60.9
5	*Aloe vera* (L.) Burm. f. JP–385	Asphodelaceae	Kawar patha	Leaves	Extract, oil	**Skin care, pimples, cuts**, constipation, bleeding gums, diabetes	0.055	0.64	87.5
6	*Amaranthus viridis* Hook JP–371	Amaranthaceae	Chulai	Whole plant	Extract, powder	Diuretic, laxative, **blood purification**, epilepsy, Cooking, fodder	0.057	0.53	71.7
7	*Amaranthus spinosus* L. JP–378	Amaranthaceae	Kurand	Whole plant	Extract, paste	**Snakebite**, laxative, diarrhea, mouth ulcer, vaginal discharge, wound healing, fever, cooking, fodder	0.088	0.51	74.5
8	*Lysimachia arvensis* var. caerulea (L.) Turland & Bergmeier JP–443	Primulaceae	Matri	Whole plant	Extract	Diuretic, **laxative**, skincare, liver disorders	0.077	0.26	61.5
9	*Avena sativa* L. JP–467	Poaceae	Jaii	Leaves, stem,	Extract, powder, paste	Diabetes, constipation, **sexual improvement**, brain health, normalize blood pressure, cooking	0.073	0.41	62.2
10	*Azadirachta indica* A. Juss. JP–489	Meliaceae	Nemb	Leaves, bark, roots, oil	Extract, oil, paste, raw	**Blood purification**, skincare, eczema, hair growth, ulcer, diabetes, fever, gums care, birth control, and abortion, antimicrobial	0.055	0.91	93.4
11	*Brassica campestris* L. JP–399	Brassicaceae	Saag	Whole plant and oil	Extract, paste	Diuretic, **milk production in mammals**, fodder, cooking	0.037	0.53	89.7
12	*Brassica nigra* L. JP–400	Brassicaceae	Ussu	Whole plant	Extract, paste, decoction	**Rheumatism**, skin care, snake bite, laxative, cooking	0.051	0.49	46.9
13	*Calotropis procera* (Aiti) Aiti. JP–382	Apocynaceae	Aakda	Leaves, sap milk	Extract, decoction	**Sexual disorders**, anti-inflammatory, vermifuge, diarrhea, analgesic	0.051	0.49	50.0
14	*Chenopodium album* L. JP–405	Chenopodiaceae	Bathoo	Whole plant	Extract, paste, decoction	Rheumatism, **diuretic**, laxative, skincare, cooking, fodder	0.043	0.69	79.0
15	*Chenopodium murale* L. JP–403	Chenopodiaceae	Jangli bathoo	Whole plant	Extract, paste, decoction	Piles, asthma, sexual disorder, cough, hair loss, **blood purification**, cooking, fodder	0.075	0.53	67.9
16	*Citrus limon* (L.) Osbeck JP–433	Rutaceae	Nimbo/Leemo	Fruit, stem bark, oil,	Extract, oil, paste	Scurvy, flu, stomach imbalance, kidney stones, sore throat, malaria, flavor, **cooling juice in summer**	0.048	0.82	74.5
17	*Convolvulus arvensis* L. JP–401	Convolvulaceae	Bhanweri	Leaves, flowers, roots	Tea, extract, paste	Laxative, **mosquito bite**, normalized menstruation, wound healing	0.070	0.28	47.4
18	*Conyza canadensis* (L.) Cronquist JP–395	Asteraceae	Bonkhady	Whole plant	Extract, oil	Rheumatism, **diarrhea**, fodder	0.58	0.26	55.8
19	*Coriandrum sativum* L. JP–389	Apiaceae	Dhanyaa	Whole plant	Extract, oil, raw	**Rheumatism**, nausea, diarrhea, intestinal gas, toothache, flavor	0.039	0.76	34.2
20	*Cordia dichotoma* Forst. f. JP–391	Boraginaceae	Lasooda/ Nasooda	Leaves, seeds, bark, fruit	Decoction, extract, paste	Fever, **anti-inflammatory**, diarrhea, headache, Pickle, cooking, fodder	0.047	0.74	64.2
21	*Coronopus didymus* (L.)Sm. JP–394	Brassicaceae	Jungle podia	Whole plant	Extract	Rheumatism, asthma, diarrhea, **malaria**, cooking, fodder	0.079	0.38	38.2
22	*Cucurbita pepo* L. JP–375	Cucurbitaceae	Loki	Leaves, flowers, fruits, seeds,	Extract, paste, raw	Diuretic, diabetes, ulcer, laxative, constipation, wound healing, **vermifuge**, cooking	0.095	0.42	51.8
23	*Cuscuta reflexa* Roxb. JP–404	Convolvulaceae	Amar beel	Whole plant/vine	Extract	Rheumatism, diuretic, **jaundice**, cough, skincare, fever, laxative	0.58	0.06	33.3
24	*Chloris virgata* Sw. JP–473	Poaceae		Leaves	Paste, extraction	Wound healing, **Fodder**	0.25	0.04	25.0
25	*Cynodon dactylon* (L.) Pers. JP–468	Poaceae	Ghass	Whole plant	Extract, powder	Skincare, laxative, **fodder**	0.068	0.22	90.9
26	*Dalbergia sissoo* Roxb. ex DC. JP–411	Fabaceae	Talii	Leaves, wood, bark	Paste, oil	**Diuretic**, diabetes, vaginal infection, sperm production, piles, nausea	0.30	0.10	35.0
27	*Datura innoxia* Mill JP–439	Solanaceae	Sundu	Fruit, seeds,	Extract, paste	**Malaria**, ulcer, common cold, asthma, piles, skin problems, hair loss, dandruff, sexual disorders,	0.35	0.13	38.5
28	*Eclipta alba* (L.) Hassk. JP–397	Asteraceae	Gokdi	Whole plant	Decoction	Rheumatism, asthma, **diabetes**, skincare, laxative, liver disorder, eye drops, hair growth	0.33	0.12	45.8
29	*Eclipta prostrate* (L.) L. JP–381	Asteraceae	Kehraj	Whole plant	Extract, paste	Liver disorders, **hair dye**, skin care, malaria, anti-inflammatory, diabetes	0.23	0.13	38.5
30	*Eucalyptus globulus* Labill. JP–465	Myrtaceae	Sufaida	Leaves, oil, seed	Extract, oil, paste	Antiseptic, vermifuge, cough, **wound healing**	0.15	0.13	48.1
31	*Ficus bengalense* (Roxb.) Wight & Arn. JP–461	Moraceae	Bohard	Leaves, hanging roots, fruit, bark	Extract, decoction	Rheumatism, vaginal infection, diabetes, diarrhea, **jaundice**, anti-inflammatory, ulcer, skincare	0.045	0.89	84.3
32	*Ficus religiosa* (L.) Forssk. JP–459	Moraceae	Peppaal	Leaves, bark, fruit	Extract, decoction, paste	Rheumatism, diabetes, jaundice, ulcer, constipation, cough, fever, wound healing, **leucorrhea**, skincare, piles	0.064	0.86	71.5
33	*Foeniculum vulgare* Mill. JP–369	Apiaceae	Saunf	Leaves, flowers, fruit	Extract, decoction, tea	Diuretic, **laxative**, asthma, acne, blood purification	0.038	0.66	66.7
34	*Hordeum vulgare* L. JP–466	Poaceae	Joo	Whole plant	Decoction, paste	Rheumatism, sore throat, **cough**, vaginal infection, constipation, cooking	0.048	0.63	78.6
35	*Lawsonia Inermis* L. JP–419	Lythraceae	Mehndi	Leaves, flowers, bark	Paste or powder, extract, decoction	Diabetes, skincare, wound healing, ornamentation (**hair dye, staining of hair, hands, legs, and nails**)	0.061	0.33	89.4
36	*Mangifera indica* L. JP–428	Anacardiaceae	Aamb/Aam	Leaves, flowers, fruits, seed, bark, stem, roots	Powder, extract, raw	skincare, diarrhea, piles, diuretic, diabetes, cardiac disorders, asthma, cough, **stomach imbalance**, cooking, pickle	0.064	0.85	81.3
37	*Melia azedarach* L. JP–451	Meliaceae	Bkain	Leaves, root, bark, gum	Extract	Asthma, diuretic, diarrhea, laxative, **emetic**, cooking	0.040	0.74	75.8
38	*Melilotus indicus* (L.) All. JP–409	Fabaceae	Sinji	Whole plant	Extract	Laxative, **diabetes**, diarrhea, cooking, fodder	0.14	0.16	59.4
39	*Mentha longifolia* (L.) Huds. JP–417	Lamiaceae	Jangli podia	Whole plant	Paste, extract, tea	**Asthma**, common cold, cough, stomach imbalance, headache, wound healing, laxative	0.065	0.54	70.4
40	*Moringa oleifera* Lam. JP–470	Moringaceae	Sohanjna	Leaves, flowers, seed, and oil	Raw, extract, paste	Diuretic, antimicrobial, asthma, diabetes, birth control, **constipation**, hair growth, diarrhea, snake bite, ulcer, wound healing, cooking	0.15	0.39	65.8
41	*Morus alba* L. JP–457	Moraceae	Shatoot	Leaves, bark, fruit	Extract, paste, raw	Cough, common cold, eye infection, **sore throat**, fever, headache	0.048	0.62	
42	*Nicotiana plumbaginifolia* L. JP–434	Solanaceae	Jangli tambaco	Leaves, stem, fruit	Extract, powder	**Rheumatism**, snake bite, wound healing, leaches removal, toothache	0.037	0.67	85.1
43	*Ocimum basilicum* L. JP–415	Lamiaceae	Niazbo	Leaves, roots, seeds,	Extract	Headache, **menstrual cramps**, diuretic, diarrhea, constipation, loss of appetite, flavor	0.041	0.86	68.6
44	*Ocimum sanctum* L. JP–413	Lamiaceae	Tulsi	The whole plant, oil	Extract, powder, oil	**Rheumatism**, asthma, cough, flu, sore throat, malaria, skincare, diabetes, diarrhea, fever, antiseptic	0.081	0.68	61.8
45	*Oxalis corniculata* L. JP–463	Oxalidaceae	Junlgi booti	Whole plant	Extract,	Rheumatism, diuretic, fever, diarrhea, flu, **snake bite**, Cooking, flavor	0.28	0.14	35.7
46	*Phragmites karka* (Retz.) Trin. Ex steud. JP–436	Poaceae	Lundda	Stem wood	Extract	Cholera, **cardiac disorders**, fish poisoning, Cooking, fodder	0.13	0.19	34.2
47	*Ricinus communis* L. JP–402	Euphorbiaceae	Arand	Leaves, roots, seeds, oil	Extract, paste	Rheumatism, **laxative**, liver disorders, castor oil for eye care, birth control, skincare	0.059	0.51	80.4
48	*Salvodora oleoides* Decne JP–430	Salvadoraceae	Jaal	Leaves, bark, roots, seed, and oil	Raw, paste, extract	Rheumatism, diuretic, **stomach imbalance**, cough, fever, kidney stones, toothache, mosquito repellent, cooking	0.069	0.65	57.7
49	*Sisymbrium irio* L. JP–393	Brassicaceae	Farmi saag	Leaves, seeds, flowers	Extract, powder, paste	Rheumatism, cough, **wound healing**, sore throat	0.091	0.22	40.9
50	*Solanum nigrum* L. JP–453	Solanaceae	Bambool	Leaves, stem	Extract, raw	Skin problems, loss of appetite, asthma, **ulcer**, laxative, cough, fever, cooking	0.069	0.58	75.9
51	*Solanum surattense* L. JP–445	Solanaceae	Rubdi	Whole plant	Decoction, paste	Rheumatism, **sore throat**, cough, vaginal infection, constipation	0.12	0.21	76.2
52	*Sonchus arvensis* L. JP–380	Asteraceae	Maii bodi	Leaves, roots, flowers	Extract	**Asthma**, cough, chest pain, caked breast treatment	0.10	0.19	41.0
53	*Syzygium cumini* L. JP–462	Myrtaceae	Jaman	Leaves, flowers, wood	Extract, raw	Rheumatism, sore throat, asthma, **blood purification**, ulcer	0.05	0.50	72.0
54	*Terminalia arjuna* L. JP–408	Combretaceae	Arjun	Leaves, bark, fruit, flower	Extract, paste, powder	**Diabetes**, cardiac disorders, cough, sore throat, diarrhea, constipation, sore eyes	0.076	0.46	71.7
55	*Triticum aestivum* L. JP–476	Poaceae	Daany/Gandam	Whole plant	Extract, paste	Diuretic, laxative, kidney problems, sexual strength in both sexes, sore throat, constipation, **cooking**	0.037	0.95	88.9
56	*Withania somnifera* (L.) Dunal JP–441	Solanaceae	Aak-Singh	Leaves, flowers, fruits	Extract	Rheumatism, **improvement of sexual strength in male**	0.032	0.31	67.7
57	*Xanthium strumarium* L. JP–432	Asteraceae	Muhabat booti	Leaves, roots, fruit	Decoction	Diuretic, laxative, improve appetite, sedative, malaria, **fever**	0.37	0.08	37.5
58	*Ziziphus mauritiana* Lam. JP–440	Rhamnaceae	Berry	Roots, leaves, fruits,	Extract, paste, oil, raw	Diuretic, **diabetes**, fever, cough, constipation, sleeplessness, liver disorders, ulcer, wound healing, skincare	0.082	0.55	68.2

* The numbers after JP represent the voucher number issued while submitting the collected specimen to the Sultan Ahmad Herbarium, Government College University Lahore. Bold letters in the “Uses” column represent the uses on which the fidelity level is based; FL %—fidelity level; RFC—relative frequency citation; UV—use value.

## Data Availability

Not applicable.

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
