# Peer review of "Quantitative Ethnobotanical Analysis of Medicinal Plants of High-Temperature Areas of Southern Punjab, Pakistan"

_plants, 2021, doi:10.3390/plants10101974_

Round 1

Reviewer 1 Report

The authors propose a manuscript titled “Quantitative ethnobotanical analysis of medicinal plants of high-temperature areas of Southern Punjab, Pakistan”. The authors discuss that the lack of proper infrastructure and poor economic conditions of rural communities make them dependent on herbal medicines. In this way is important to obtain and conserve the historic and traditional knowledge about the medicinal importance of different plants used by the inhabitants of very long historic villages as in Southern Punjab, Pakistan. In particular the study schows results on 58 plant wild species based on 200 experienced respondents. The Poaceae family resulted the most predominant family, followed by Solanaceae and Asteraceae. Many details are documented in the manuscript as about which part of the plant were utilized, in which way, their properties. The manuscript is original in the data compared to other similar articles and is able to be published on international audience as Plants. However I believe it is necessary to implement the maniscript with some crucial concepts that the authors will have no problem to accepting as they are designed to improve the work.

Abstract

  • Summarize the abstract without detailed data

Introduction

Well done, few suggestions.

  • “….Ethnobotanical surveys are often conducted to evaluate the complex connection between local communities and wild plants [4]”. I think that the authors want to refer on wild species not cultivated one.
  • The author statement “Traditional ethnobotanical data of medicinal plants of decades is required for the protection, conservation, and development of herbal drugs”. True, especially for endemic or endangered species. Two more about this concept if in the species considered there are species with these characteristics.
  • Since ancient times, humans have started extracting and processing valuable materials from medicinal plant species to cure and treat several diseases” (Choose a reference).
  1. Materials and Methods

Few observations.

  • The authors statement: “Plants were collected, and all the necessary data was documented, including their vernacular names, locality, and medicinal values. Plants were identified with the assistance of taxonomic experts available at the Department of the Botany, GC University La- hore, and available literature present in the Science Library of the University [24,25]. Moreover, the plants were also compared with the existing plants available at the university’s herbarium, Sultan Ahmad Herbarium. The collected specimens were pressed, dry out, and deposited to Sultan Ahmad Herbarium”. The samples were collected and deposited in an official herbarium? if yes, please indicate the code of each samples and the name of the latin herbarium name
  1. Discussion and results

Well done. Few observations

  • Please complete the scientific name with the author that discovered for the first time the species. Check whole document in this way.
  • Table 3, column 2. Please change voucher in specimen
  • Check the updated scientific name in whole document. E.g. Lysimachia arvensis (L.) U. Manns & Anderb. and not the old name Anagallis arvensis
  1. Conclusion

Please complete the last period in the suggested way: “…In conclusion, there were various plants in the study area, which had great ethnobotanical potential to treat various diseases as revealed through different indices. In the future it is hoped to identify in the same study area with the help of field botanists, further species potentially useful to humans, from medicinal plants [Valerio et al. 2021, Perrino et al. 2021] to Crop Wild Relatives [Perrino and Perrino 2020, Maxted et al. 2011, Maxted and Kell 2009] that have recently registered the attention of the global scientific community.

References to be added

  • Valerio, F.; Mezzapesa, G.N.; Ghannouchi, A.; Mondelli, D.; Logrieco, A.F.; Perrino, E.V. Characterization and antimicrobial properties of essential oils from four wild taxa of Lamiaceae family growing in Apulia. Agronomy 2021, 11, 1431. https://doi.org/10.3390/agronomy11071431
  • Perrino, E.V.; Valerio, F.; Gannouchi, A.; Trani, A.; Mezzapesa, G. Ecological and Plant Community Implication on Essential Oils Composition in Useful Wild Officinal Species: A Pilot Case Study in Apulia (Italy). Plants 2021, 10, 574. https://doi.org/10.3390/plants10030574
  • Perrino, E.V., Perrino, P. Crop wild relatives: know how past and present to improve future research, conservation and utilization strategies, especially in Italy: a review. Resour. Crop Evol. 2020, 67, 1067–1105. https://doi.org/10.1007/s10722-020-00930-7
  • Maxted, N.; Dulloo, M.E.; Ford-Lloid, B.V.; Frese, L.; Iriondo, J.; Pinheiro de Carvallho, M.A.A. Agrobiodiversity conservation securing the diversity of crop wild relatives and landraces. CAB International, Wallingford, 2011.
  • Maxted, N.; Kell, S. Establishment of a global network for the in situ conservation of crop wild relatives: status and needs. Study Pap. 2009, 39, 1–224

References

Please follow the guidelines of the journal and correct.

Author Response

Response to the Reviewer 1 comments

The authors propose a manuscript titled “Quantitative ethnobotanical analysis of medicinal plants of high-temperature areas of Southern Punjab, Pakistan”. The authors discuss that the lack of proper infrastructure and poor economic conditions of rural communities make them dependent on herbal medicines. In this way is important to obtain and conserve the historic and traditional knowledge about the medicinal importance of different plants used by the inhabitants of very long historic villages as in Southern Punjab, Pakistan. In particular, the study shows results on 58 plant wild species based on 200 experienced respondents. The Poaceae family resulted the most predominant family, followed by Solanaceae and Asteraceae. Many details are documented in the manuscript as about which part of the plant were utilized, in which way, their properties. The manuscript is original in the data compared to other similar articles and is able to be published on international audience as Plants. However, I believe it is necessary to implement the manuscript with some crucial concepts that the authors will have no problem to accepting as they are designed to improve the work.

Abstract

Summarize the abstract without detailed data

Response: We have tried our level best to remove the unnecessary sentence/s and summarized the main results recorded during the present study.

Introduction

Well done, few suggestions.

Response: Thanks for your appreciation

“….Ethnobotanical surveys are often conducted to evaluate the complex connection between local communities and wild plants [4]”. I think that the authors want to refer on wild species not cultivated one.

Response: We agree with the reviewer. We have replaced “wild plants” with “wild species of plants” (Please see line 55)

The author statement “Traditional ethnobotanical data of medicinal plants of decades is required for the protection, conservation, and development of herbal drugs”. True, especially for endemic or endangered species. Two more about this concept if in the species considered there are species with these characteristics.

Response: Since the present research work, the species found in the study area are not among the ones, which are endangered. Therefore, we have not made this sentence not focused but a general one 

“Since ancient times, humans have started extracting and processing valuable materials from medicinal plant species to cure and treat several diseases” (Choose a reference).

Response: We have inserted the reference for the mentioned sentence (Please see line 65)

Materials and Methods

Few observations.

The authors statement: “Plants were collected, and all the necessary data was documented, including their vernacular names, locality, and medicinal values. Plants were identified with the assistance of taxonomic experts available at the Department of the Botany, GC University Lahore, and available literature present in the Science Library of the University [24,25]. Moreover, the plants were also compared with the existing plants available at the university’s herbarium, Sultan Ahmad Herbarium. The collected specimens were pressed, dry out, and deposited to Sultan Ahmad Herbarium”. The samples were collected and deposited in an official herbarium? if yes, please indicate the code of each samples and the name of the latin herbarium name

Response: We have already added the code of each species collected and deposited in the, Sultan Ahmad Herbarium at Government College University Lahore. These code start with “JP” (Please see Table 4)

Discussion and results

Well done. Few observations

Response: Thanks for your appreciation

Please complete the scientific name with the author that discovered for the first time the species. Check whole document in this way.

Response: We have crosschecked all the italic names of species, corrected where required and highlighted as yellow

Table 3, column 2. Please change voucher in specimen

Response: As per suggestion of the reviewer, we have changed “voucher” with “specimen”

Check the updated scientific name in whole document. E.g. Lysimachia arvensis (L.) U. Manns & Anderb. and not the old name Anagallis arvensis

Response: We have crosschecked all the names of the plants mentioned in Table 4 and corrected where required

Conclusion

Please complete the last period in the suggested way: “…In conclusion, there were various plants in the study area, which had great ethnobotanical potential to treat various diseases as revealed through different indices. In the future it is hoped to identify in the same study area with the help of field botanists, further species potentially useful to humans, from medicinal plants [Valerio et al. 2021, Perrino et al. 2021] to Crop Wild Relatives [Perrino and Perrino 2020, Maxted et al. 2011, Maxted and Kell 2009] that have recently registered the attention of the global scientific community.

References to be added

Valerio, F.; Mezzapesa, G.N.; Ghannouchi, A.; Mondelli, D.; Logrieco, A.F.; Perrino, E.V. Characterization and antimicrobial properties of essential oils from four wild taxa of Lamiaceae family growing in Apulia. Agronomy 2021, 11, 1431. https://doi.org/10.3390/agronomy11071431

Perrino, E.V.; Valerio, F.; Gannouchi, A.; Trani, A.; Mezzapesa, G. Ecological and Plant Community Implication on Essential Oils Composition in Useful Wild Officinal Species: A Pilot Case Study in Apulia (Italy). Plants 2021, 10, 574. https://doi.org/10.3390/plants10030574

Perrino, E.V., Perrino, P. Crop wild relatives: know how past and present to improve future research, conservation and utilization strategies, especially in Italy: a review. Resour. Crop Evol. 2020, 67, 1067–1105. https://doi.org/10.1007/s10722-020-00930-7

Maxted, N.; Dulloo, M.E.; Ford-Lloid, B.V.; Frese, L.; Iriondo, J.; Pinheiro de Carvallho, M.A.A. Agrobiodiversity conservation securing the diversity of crop wild relatives and landraces. CAB International, Wallingford, 2011.

Maxted, N.; Kell, S. Establishment of a global network for the in situ conservation of crop wild relatives: status and needs. Study Pap. 2009, 39, 1–224

Response: We have consulted the suggested references and cited as per suggestion of the reviewer

References

Please follow the guidelines of the journal and correct.

Response: We have crosschecked all the references and formatted as per for the format of the journal

Reviewer 2 Report

This paper report the ethnobotanical analysis of medicinal plants of 
Southern Punjab, especially Saraiki language speaking area. If this paper is the first report for the survey of Saraiki language speaking people, please explain it and emphasize the importance of this paper, and explain more about new finding, new usage of medicinal plants in this region. 

Most of the plants in Table 3 are common to other area or cosmopolitan. Please focus on the domestic medicinal plants in these region and explain about the importance of them.

Other comments are in the attached file with colour markers and comments. Please check and revise as much as possible.

Author Response

Response to the reviewer 2 comments

Means not so old....In Pakistan, these villages are very old? In other countries, three hundred years are not so old. Many city in China, Korea, Japan, Vietnam, etc has history of more than 1,000.

Better to change explanation why you select these villages.

Response: These villages are comparatively remote and situated near the bank of River Ravi. Moreover, these villages do not have modern healthcare centers and people for centuries are relying on medicinal plants. Unfortunately, no research work has been conducted to document the ethnomedicinal knowledge of plants found in these villages in the past (Please see lines 97-101)

This paper report the ethnobotanical analysis of medicinal plants of Southern Punjab, especially Saraiki language speaking area. If this paper is the first report for the survey of Saraiki language speaking people, please explain it and emphasize the importance of this paper, and explain more about new finding, new usage of medicinal plants in this region.

Please explain this. What kind of people, history, and background related to medicinal plants?

Response: The Saraiki language is majorly spoken in Southern Punjab, Pakistan. This language has emerged from several dialects after the independence of Pakistan in 1947. Saraikstan is a combined region combined of different areas where this language is spoken.  Saraiki language people are considered friendly and known for their hospitality. Some 40,000 years back, the Saraiki region became part of the Indus Valley Civilization. There are about 8.38% of people in Pakistan, who spoke the Saraiki language. Most of the Saraiki language people are living in the rural areas and healthcare centers are far away from their reach. Therefore, these people have remained more inclined towards the use of medicinal plants for centuries (Gilani, 2013). (Please see lines 105-114)

Reference

Gilani, M.H., 2013. Historical Background of Saraiki Language. Pak. J. Soc. Sci. 33(1), 61-76

So many illiterate! Can you assure the reliability of this research?

Response: Ethnobotanical knowledge usually passes from generation to generation verbally and most of the people from Pakistan have expertise in using medicinal plants despite the high rate of illiteracy. It is a common trend that most of the older people will not even write a single sentence but they are genius in using medicinal plants and are famous for their knowledge about the medicinal plants in the surrounding areas. Therefore, these people have a bundle of knowledge about the use of medicinal plants despite being illiterate (Please see lines 127-133)

Most of the plants in Table 3 are common to other area or cosmopolitan. Please focus on the domestic medicinal plants in this region and explain about the importance of them.

Response: The present research work is novel compared to ethnomedicinal literature of other areas of Southern Punjab (3 number reference in the manuscript; Ahmed et al., 2015; Badar et al., 2017). The results have been only compared to ethnomedicinal literature of neighboring areas because the distant areas have fewer similarities. Comparative studies have revealed that twenty plant species including Albizia lebbeck, Amaranthus spinosus, Brassica nigra, Chenopodium murale, Conyza Canadensis, Cordia dichotoma, Coronopus didymus, Cucurbita pepo, Eclipta alba, Eclipta prostrate, Eucalyptus globules, Hordeum vulgare, Lawsonia Inermis, Ocimum basilicum, Ocimum sanctum, Ricinus communis, Salvodora oleoides, Sisymbrium irio, Sonchus arvensis, and Terminalia arjuna were not documented previously. In the present study, newly reported species with their most common uses for the first time include Albizia lebbeck (Diarrhea), Amaranthus spinosus (Snake bite), Brassica nigra (Rheumatism), Chenopodium murale (blood purification), Conyza Canadensis (Diarrhea), Cordia dichotoma (Anti-inflammatory), Coronopus didymus (malaria), Cucurbita pepo (vermifuge), Eclipta alba (diabetes), Eclipta prostrate (hair dye), Eucalyptus globules (Wound healing), Hordeum vulgare (Cough), Lawsonia Inermis (Hair dye, staining of hair, hands, legs, and nails), Ocimum basilicum (Menstrual cramps), Ocimum sanctum (Rheumatism), Ricinus communis (Laxative), Salvodora oleoides (Stomach imbalance), Sisymbrium irio (Wound healing), Sonchus arvensis (Asthma), and Terminalia arjuna (Diabetes).

The present study also reported that nineteen species could be used for the treatment of rheumatism. In literature, no study has reported such a good number of species for the treatment of rheumatism. On the other hand, sixteen species have been documented for the treatment of diarrhea and these results are pivotal for the researchers to focus on the detection of useful bioactive compounds for the preparation of novel drugs.

References

Badar, N., Iqbal, Z., Sajid, M.S., Rizwan, H.M., Jabbar, A., Babar, W., Khan, M.N. and Ahmed, A., 2017. Documentation of ethnoveterinary practices in district Jhang, Pakistan. J Anim Plant Sci, 27(2), pp.398-406.

Ahmed, N., Mahmood, A., Mahmood, A., Sadeghi, Z. and Farman, M., 2015. Ethnopharmacological importance of medicinal flora from the district of Vehari, Punjab province, Pakistan. Journal of ethnopharmacology, 168, pp.66-78.

We have explained about the plants, which were novel to the study area (Please see line 357-378)

Other comments are in the attached file with color markers and comments. Please check and revise as much as possible.

Response: We have checked the comments, corrected them as per the comments/suggestions of the reviewer, and highlighted them as yellow

As per the suggestion of the reviewer, we have presented the data in Figure 3 as Table and modified Figures 3-5 and Table 3 and 4 as per comments of the reviewer

Round 2

Reviewer 1 Report

The authors modified the original manuscript following all suggestions of the reviewer. 

The last version is able to be published on the journal

Author Response

Response to the Reviewer 1 comments

The authors modified the original manuscript following all suggestions of the reviewer. The last version is able to be published on the journal

Response: Thanks for your recommendation

Reviewer 2 Report

You revised very well according to my comments and suggestions. Only one mistake and one question.

1) At Table 4 and your answer to me, "Conyza Canadensis --> Conyza canadensis.

2) At Table 4, please add explanation of "JP-454, etc". It is better to add explanation under this Table. In my opinion, this number is not necessary for this paper. No meaning for the readers, then I think it is better to delete these number. 

Author Response

Response to the reviewer 2 comments

You revised very well according to my comments and suggestions. Only one mistake and one question.

Response: Thanks for your appreciation. We have incorporated your suggestions in the revised manuscript

1) At Table 4 and your answer to me, "Conyza Canadensis --> Conyza canadensis.

Response: Thanks for pointing out this mistake. We have corrected it and highlighted it as yellow in Table 4 in the revised manuscript

2) At Table 4, please add explanation of "JP-454, etc". It is better to add explanation under this Table. In my opinion, this number is not necessary for this paper. No meaning for the readers, then I think it is better to delete these number.

Response: “The numbers after JP represent the voucher number issued while submitting the collected specimen in Sultan Ahmad Herbarium, Government College University Lahore” We have added this statement as a footnote of Table 4 as an explanation for "JP-454, etc"

The main reason for their addition in Table 4 is that these are the number that could be verified at the herbarium for their future authenticity and further research

In literature, many papers have used this type of voucher number e.g. http://dx.doi.org/10.1016/j.hermed.2017.09.004

http://dx.doi.org/10.1016/j.hermed.2017.08.001

http://dx.doi.org/10.1016/j.hermed.2017.04.002

https://doi.org/10.1016/j.hermed.2019.100271
